# RETSim: Resilient and Efficient Text Similarity

**Marina Zhang[1], Owen Vallis[1], Aysegul Bumin[*2], Tanay Vakharia[1], Elie Bursztein[1]**
Google[1]    University of Florida[2]

## Abstract

This paper introduces RETSim (Resilient and Efficient Text Similarity), a lightweight, multilingual deep learning model trained to produce robust metric embeddings for near-duplicate text retrieval, clustering, and dataset deduplication tasks. We demonstrate that RETSim is significantly more robust and accurate than MinHash and neural text embeddings, achieving new state-of-the-art performance on dataset deduplication, adversarial text retrieval benchmarks, and spam clustering tasks. We also introduce the W4NT3D benchmark (Wiki-40B 4dversarial Near-T3xt Dataset) for evaluating multilingual, near-duplicate text retrieval capabilities under adversarial settings. RETSim and the W4NT3D benchmark are released under the MIT License at https://github.com/google/unisim.

## 1 Introduction

Robust near-duplicate text detection is an essential component of many tasks, including retrieving documents, detecting plagiarism (Sun et al., 2013) and blocking adversarial spam campaigns (Ahmed et al., 2022). Users have come to expect that systems can return accurate results despite their queries exhibiting a 20% to 30% typo rate (Hagen et al., 2017). Furthermore, efficiently deduplicating text datasets is critical to training state-of-the-art large language models (Lee et al., 2022; Kandpal et al., 2022).

For more than two decades, MinHash-based (Broder et al., 1998) locality-sensitive hashing (LSH) has been the most prevalent algorithm used for near-duplicate detection due to its simplicity, robustness, and speed. For example, the vast majority of dataset deduplication efforts still rely on MinHash (Lee et al., 2022; Kocetkov et al., 2022). However, like all LSH-based techniques, MinHash is not without downsides; chief among them being that it is very parameter-sensitive and requires heavy tuning. Additionally, MinHash lacks resilience to typos due to its reliance on n-grams, leading to poor performance on noisy data and a vulnerability to hash-busting attacks (Issac et al., 2014).

On the other hand, deep learning models are the dominant way to perform vector-based semantic text retrieval (Muennighoff et al., 2022), but so far, no neural embedding has been able to consistently outperform MinHash for robust near-duplicate detection (Silcock et al., 2022). This is mostly due to the focus on improving semantic capabilities, which leads models to be too large to run extremely quickly and the use of sub-word level tokenization, which is not resilient to typos and adversarial attacks (Morris et al., 2020; Bursztein et al., 2023).

To fill this gap, we introduce RETSim (Resilient and Efficient Text Similarity), a lightweight, multilingual deep learning model trained specifically to produce robust neural embeddings specialized for near-duplicate detection. By combining the state-of-the-art RETVec text vectorizer, a modern transformer block (Hua et al., 2022), a large typo-augmented training corpus, and a metric learning training regime, RETSim is able to achieve new state-of-the-art performance on near-duplicate detection benchmarks (Section 4.2), dataset deduplication tasks (Sections 4.3 and 5.1), and spam clustering applications (Section 5.2).

Furthermore, while datasets and benchmarks exist for corpus deduplication and near-duplicate text retrieval, none of these have focused on systematically evaluating near-duplicate retrieval performance under the presence of typos, word manipulations, and sentence or paragraph-level modifica-

---

[*]This work was done during the author's internship at Google.

tions. To address this need, we additionally introduce the *W4NT3D* benchmark (Wiki-40B 4dversarial Near-T3xt Dataset) which enables the evaluation of algorithms on adversarial near-duplicate text retrieval in a multilingual setting. We report the performance of RETSim, MinHash, and popular neural embeddings such as Universal Sentence Encoder (Cer et al., 2018) and LaBSE (Feng et al., 2022) on this new benchmark in Section 4.2, highlighting uneven performance across languages and types of adversarial manipulations. The RETSim model and the W4NT3D benchmark are made available at https://github.com/google/unisim under the MIT License.

## 2 RELATED WORK

**Near-Duplicate Detection**    Identifying noisy near-duplicate documents in a large corpus is a fundamental task with a wide range of applications, such as detecting plagiarism, finding reproduced content in literature or news articles (Gyawali et al., 2020; Silcock et al., 2022), and deduplicating training datasets for language models. Previous research has shown that duplicates in training datasets lead to inefficient training (Lee et al., 2022) and privacy concerns for large language models (LLMs), where models memorize and regenerate duplicated training sequences at a much higher frequency (Kandpal et al., 2022).

Unlike semantic text similarity, the task of identifying textual near-duplicates has been predominated by non-neural, n-gram-based algorithms such as MinHash (Broder et al., 1998), which is the most widely used technique for deduplicating large training corpuses (Kocetkov et al., 2022; Lee et al., 2022). MinHash is a technique for estimating the Jaccard similarity between two sets. Algorithms such as MinHash or SimHash (Charikar, 2002) can be combined with locality-sensitive hashing (LSH) techniques for fast, approximate nearest neighbor search and data clustering. This allows them to scale and deduplicate corpuses containing terabytes of data such as C4 (Lee et al., 2022) and The Stack (Kocetkov et al., 2022). However, n-gram or shingling-based techniques typically require texts to be parsed into a standardized form (e.g. by lower-casing or stripping punctuation), which makes them susceptible to typos and adversarial attacks and pose a challenge when attempting to differentiate between dissimilar documents and near-duplicate documents with adversarial augmentations.

**Semantic Text Similarity**    The task of computing semantic similarity between text is closely related to near-duplicate detection. Semantic text similarity refers to the assessment of the semantic relatedness of two pieces of text based on their meaning rather than their syntactic structure, as in the case of near-duplicate detection. Recently, transformer-based language models such as Universal Sentence Encoder (Yang et al., 2019), LaBSE (Feng et al., 2022) and LLM-based embeddings (Anil et al., 2023) which embed text into high-dimensional embedding vectors have been successfully used to retrieve semantically-related documents using cosine similarity. Modern text retrieval systems combine these embeddings with an approximate nearest neighbor (ANN) search algorithm to efficiently retrieve documents matching user queries.

However, language models have been shown to be vulnerable to adversarial attacks and naturally-occurring typos (Alzantot et al., 2018; Gao et al., 2018; Morris et al., 2020). Furthermore, language models are typically very large and costly to run even with hardware acceleration, which makes them unsuited for large-scale dataset deduplication or identifying near-duplicates in the presence of typos or adversarial text manipulations.

**Metric Learning**    Metric learning aims to learn an embedding space where similar items have a small distance between their embeddings and dissimilar items are further away. Many state-of-the-art embeddings use metric learning for unsupervised training or fine-tuning including Sentence-BERT (Reimers & Gurevych, 2019) and E5 (Wang et al., 2022).

RETVec is a resilient, multilingual embedding and text vectorizer trained to be robust against various forms of character-level typos and adversarial attacks. We extend the RETVec training regime to full text documents for RETSim. We use Multi-Similarity Loss (Wang et al., 2019) for pair-based metric learning, where typo-laden and near-duplicate versions of texts are trained to be closer in the embedding space, while other texts are pushed further away. Multi-Similarity Loss is based on a general weighting framework for pair-based losses and achieves state-of-the-art performance, outperforming alternatives such as Triplet Loss (Schroff et al., 2015).

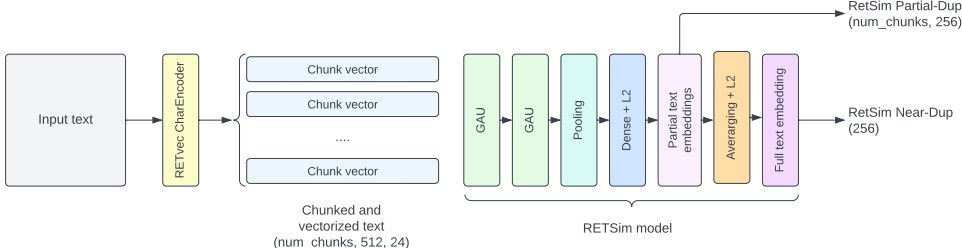

Figure 1: RETSim model architecture diagram. RETSim works on arbitrary length text by splitting texts into chunks of 512 characters during its vectorization phase and encodes them using the RETVec character vectorizer. The RETSim model then embeds each chunk of text into 256-dim partial embeddings and combines them to produce the global embedding.

## 3 RETSIM

### 3.1 ARCHITECTURE

The RETSim model is composed of three main components (as depicted in Figure 1):

**The character-level vectorizer** splits the input text into chunks of 512 characters, then uses the RETVec chararcter encoder (Bursztein et al., 2023) to encode each chunk, resulting in a batch of $(512, 24)$ dense inputs. The RETVec character vectorizer encodes each Unicode character as a compact 24-bit binary representation based on its integer codepoint value. This allows the vectorizer to encode all valid Unicode characters and support all languages. Furthermore, the character-level vectorizer has been shown to be more resilient against typos and adversarial attacks.

**A small transformer model** is used to compute 256-dimension embeddings for each chunk of the input text. RETSim$_{\text{Partial-Dup}}$ uses these embeddings directly to finding documents that have matching chunks of text. Architecturally, the model consists of two Gated Attention Unit (GAU) blocks (Hua et al., 2022), followed by a Generalized-Mean pooling layer (Radenović et al., 2018), a dense projection layer which projects the embedding into 256 dimensions, and an L2 normalization layer. The model has only 536k parameters, which is more than two orders of magnitude smaller than other neural embeddings (Table 1). L2-normalization allows the embeddings to be compared using cosine similarity. We discuss the impact of key architecture design choices in Section 6. Hyperparameter details are provided in Appendix A.1.1, and additional ablations results in Appendix A.5.

**An embedding averaging module** is then used to combine partial text embeddings into a full-text embedding which is used for global near-duplicate matching (RETSim$_{\text{Near-Dup}}$). Averaging chunked embeddings to produce a global embedding is a standard technique used by many models (Cer et al., 2018) to support infinite length inputs in a cost-efficient manner. We experimented with other aggregation techniques to produce more accurate global embeddings, including training a deep-averaging network (Iyyer et al., 2015), but this did not improve performance and resulted in higher computation cost. RETSim$_{\text{Near-Dup}}$ and RETSim$_{\text{Partial-Dup}}$ are computed in a single forward pass which makes it computationally efficient. We output both types of embeddings as they have different applications: RETSim$_{\text{Near-Dup}}$ is better-suited for full-text matching and retrieval (Section 4.2), while RETSim$_{\text{Partial-Dup}}$ is used to find partial text matches where the near-duplicate content appears only in part of the document (Section 4.3).

### 3.2 MODEL TRAINING

**Dataset** We use the multilingual C4 dataset (mC4) for raw text data and following (Xue et al., 2020), we use a language sampling exponent of $\alpha = 0.3$ to balance sampling between low and high-resource languages. We only use text containing at least 16 characters, and we randomly select between 1 and 8 sentences (roughly 512 characters) for each text chunk. For each example in the

training dataset, we generate 5 pairs of augmented examples. We apply three levels of augmentation to each example text chunk (in this order): sentence-level, word-level, and character-level. For each level, we randomly select the augmentation to be applied from the following categories: insertion, deletion, substitution, and transposition. We randomly apply between $0 - 25\%$ sentence-level augmentation and up to $30\%$ combined character and word-level augmentation. Empirically, we found that increasing the percentage of augmentation beyond this point causes RETSim's performance to degrade. The full list of augmentations used can be found in Appendix A.2.

**Training Procedure**  We train RETSim using Multi-Similarity Loss (Wang et al., 2019) with $\alpha = 4$, $\beta = 40$, $\lambda = 0.5$, and $\epsilon = 0.1$. We hypertuned these parameters and the results are shown in Appendix A.5. We train for 1 million steps with batch size $= 1024$. The similarity loss trains the model to embed augmented versions of the same text closer in the embedding space, while dissimilar texts are pushed further apart. We use the LAMB optimizer (You et al., 2019) with a max learning rate of 0.001 and cosine decay. Detailed training hyperparameters are reported in Appendix A.1.2.

## 4 EVALUATION

| Model/Algorithm | Type | Embed./Hash Size | # Model Parameters |
|---|---|---|---|
| LaBSE | Neural | 768 | 471M |
| Multilingual USE | Neural | 512 | 69M |
| Multilingual E5-Base | Neural | 768 | 278M |
| PaLM 2 (Gecko) | Neural | 768 | ? |
| SimHash | Hashing | $b$ bits | N/A |
| MinHash | Hashing | $n$ hashes | N/A |
| RETSim | Neural | 256 | 536k |

Table 1: Embedding models and hashing algorithms benchmarked in the paper.

### 4.1 MODELS AND ALGORITHMS EVALUATED

We benchmark RETSim against four multilingual semantic text embeddings as well as popular n-gram based algorithms primarily used in near-duplicate text detection (Table 1). Our baseline text embeddings include Multilingual Universal Sentence Encoder  (Yang et al., 2019), LaBSE (Feng et al., 2022), Multilingual E5 (Wang et al., 2022), and PaLM 2 Gecko Embeddings (Anil et al., 2023). All text embeddings are L2-normalized and compared using cosine similarity. We use exact search to index and retrieve nearest neighbors from our vector index for the experiments in Section 4.

For non-neural near-duplicate detection and clustering algorithms, we selected the two most popular algorithms: MinHash (Broder et al., 1998) and SimHash (Charikar, 2002). For MinHash, we use Datasketch's MinHashLSH library. Following the most common practices in the literature (Silcock et al., 2022), we use 10 hash functions for MinHash unless otherwise specified. We use word-level n-grams where we select the best value out of $n = \{2, 3, 4, ..., 10\}$. For SimHash, we use 64-bit SimHash and conduct shingling at the character level, where the shingle size is selected from $n = \{2, 3, 4, ..., 10\}$. For the near-duplicate detection benchmarks (NEWS-COPY and CORE Near-Duplicates datasets), we tune the optimal deduplication threshold (e.g. based on cosine similarity for neural-based embeddings and Jaccard similarity for MinHash). Detailed hyperparameter settings for RETSim and baseline algorithms used in the evaluation can be found in Appendix A.3.

### 4.2 W4NT3D: WIKI-40B 4DVERSARIAL NEAR-T3XT DATASET EVALUATION

**Dataset Description**  The vast majority of text retrieval benchmarks are focused on evaluating semantic performance. To the best of our knowledge, there is no multilingual benchmark for systematically measuring adversarial robustness for near-duplicate text retrieval. In an attempt to fill in the gap, we create and publish the W4NT3D benchmark (Wiki-40B 4dversarial Near-T3xt Dataset), which contains around 400k pairs of syntactically similar texts to evaluate near-duplicate text retrieval in the presence of various forms of text manipulations and typos.

W4NT3D is based on the Wiki-40B dataset (Guo et al., 2020). The dataset is split into query examples and target examples, where query examples are synthetically-modified near-duplicate versions

of a target example (e.g. with typos). For each of the 41 language splits in Wiki-40B, we randomly select 10,000 texts. The length of the target string is uniformly selected from between 16 and 8192 characters, in order to test performance on short and long text. To construct the query text corresponding to a target text, we randomly apply up to 25% word and character augmentations, and up to 25% sentence and paragraph augmentations. For each augmentation, we uniformly select from the [insert, delete, substitute, and swap] operations. We use Recall@$k$ with $k = 1$ as the main metric, following the setup commonly found in semantic text retrieval benchmarks.

| Model/Algorithm | Arabic | Chinese | English | German | French | Spanish | Japanese | Korean | Russian | Thai | Avg (41 Langs) |
|---|---|---|---|---|---|---|---|---|---|---|---|
| LaBSE | 0.915 | 0.917 | 0.944 | 0.931 | 0.930 | 0.888 | 0.931 | 0.949 | 0.918 | 0.882 | 0.921 |
| Multilingual USE | 0.915 | **0.986** | 0.958 | 0.942 | 0.938 | 0.903 | **0.990** | 0.984 | 0.910 | 0.888 | 0.912 |
| Multilingual E5-Base | 0.936 | 0.980 | 0.959 | 0.944 | 0.948 | 0.896 | 0.979 | 0.986 | 0.911 | 0.921 | 0.937 |
| PaLM 2 (Gecko) | 0.497 | 0.623 | 0.961 | 0.932 | 0.934 | 0.911 | 0.578 | 0.701 | 0.851 | 0.571 | 0.823 |
| SimHash | 0.558 | 0.276 | 0.591 | 0.561 | 0.519 | 0.513 | 0.465 | 0.593 | 0.554 | 0.669 | 0.550 |
| MinHash | 0.633 | 0.172 | 0.591 | 0.558 | 0.556 | 0.575 | 0.223 | 0.814 | 0.523 | 0.416 | 0.538 |
| RETSim$_\text{Partial-Dup}$ | 0.928 | 0.946 | 0.954 | 0.949 | 0.947 | 0.938 | 0.963 | 0.971 | 0.946 | 0.941 | 0.949 |
| RETSim$_\text{Near-Dup}$ | **0.971** | 0.971 | **0.987** | **0.978** | **0.983** | **0.976** | 0.986 | **0.991** | **0.970** | **0.946** | **0.977** |

Table 2: Per-language retrieval performance for various embedding models and algorithms on the W4NT3D benchmark. Results on selected languages are reported alongside the average Recall@1 for all 41 languages. Full results for all languages are reported in Appendix A.4.

**Multilingual Performance**   Overall, RETSim$_\text{Near-Dup}$ achieves an average Recall@1 of 0.977 across all 41 languages on the W4NT3D benchmark (Table 2). RETSim$_\text{Partial-Dup}$ is second best with a Recall@1 of 0.949 and Multilingual E5, the best-performing baseline, is third with an average Recall@1 of 0.932. We expect that RETSim$_\text{Near-Dup}$ outperforms RETSim$_\text{Partial-Dup}$ because the W4NT3D benchmark requires an algorithm to not just find near-duplicates, but to find the most similar text. RETSim$_\text{Partial-Dup}$ optimizes for finding the most similar chunk of text in the corpus, which is not always the most similar text overall. Similarly, we hypothesize that MinHash and SimHash perform poorly on the W4NT3D benchmark due to their lack of ability to distinguish which is the most similar text among the near-duplicates detected, and embedding-based models and cosine similarity offer a more fine-grained measure of similarity.

RETSim$_\text{Near-Dup}$ outperforms baseline algorithms on all languages except for Chinese and Japanese. For these languages, we theorize that semantic embeddings may have the slight edge in performance because their significantly larger model sizes (more than 100x larger than RETSim, as shown in Table 1) allow them to have a better representation on languages with large character sets. Furthermore, the sub-word level tokenizers used in the baseline embeddings often treat each character in Chinese or Japanese as individual tokens, which could offer higher resilience to typos.

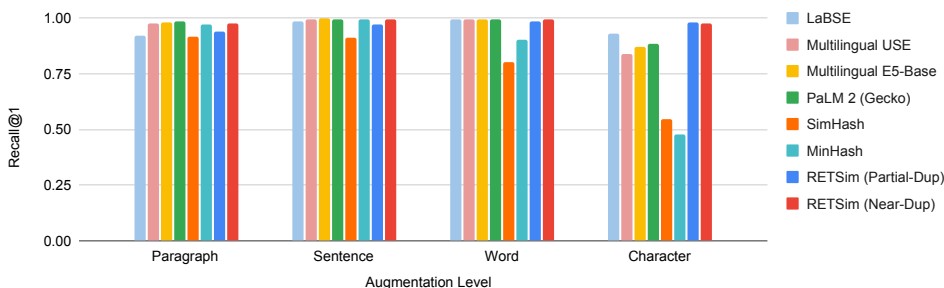

Figure 2: Recall@1 performance on the W4NT3D benchmark, broken down by augmentation type. Results are averaged across all 41 language splits in W4NT3D.

**Adversarial Resilience**   Delving deeper into the impact of various types of text manipulation reveals that RETSim$_\text{Near-Dup}$ and RETSim$_\text{Partial-Dup}$ perform almost equally well regardless of the type of augmentation applied (Figure 2). Semantic text embeddings perform well on paragraph, sentence and word-level manipulations, but as expected, exhibit significantly weaker performance towards character-level typos. MinHash and SimHash struggle more with word-level augmentations than deep-learning based embeddings and collapse when character-level typos are introduced. We at-

tribute RETSim's resilience to adversarial manipulations to the RETVec character encoder as well as using deep metric learning to train robust embeddings.

Figure 4 reports the Recall@1 performance of the algorithms as the amount of augmentation increases. All algorithms perform perfectly when no augmentation is applied (exact matching), but as the percentage of augmentation increases, n-gram based approaches exhibit a steep drop in performance. Semantic text embeddings are able to sustain a larger degree of augmentation before their retrieval capabilities start to degrade (over 20%). RETSim$_{\text{Near-Dup}}$ is the most robust algorithm, with a noticeable drop in performance only after around 40% augmentation. This makes RETSim the most effective approach at clustering and deduplicating text under adversarial settings.

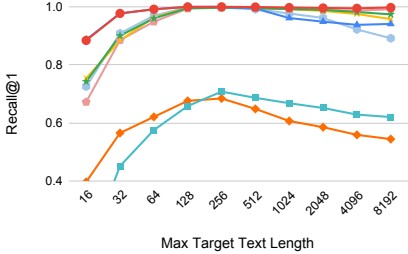
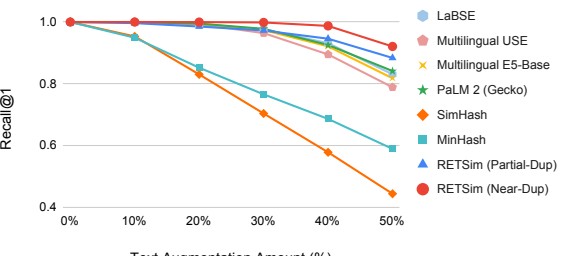

Figure 3: Recall@1 performances on the W4NT3D benchmark (English only) for varying max target lengths.

Figure 4: Recall@1 performances on the W4NT3D benchmark (English only) as the amount of augmentation applied to the query text increases.

**Text Length Impact on Performance** Figure 3 reports the Recall@1 performance of RETSim and baseline algorithms as the length of the query and target text varies. We see that RETSim$_{\text{Near-Dup}}$ and RETSim$_{\text{Partial-Dup}}$ outperforms all other methods on short texts with fewer than 128 characters. As the text length increases beyond 512 characters, RETSim$_{\text{Near-Dup}}$ remains close to perfect while RETSim$_{\text{Partial-Dup}}$'s performance degrades since it splits the text into multiple embeddings and finds the nearest matching *chunk* of text. MinHash and SimHash also perform poorly on short text lengths and start to degrade on longer texts. For neural-based embeddings, we observe a slight drop in performance on longer texts for all models except RETSim$_{\text{Near-Dup}}$ and Multilingual USE, the only two embeddings that can handle arbitrary length inputs.

## 4.3 REAL-WORLD NEAR-DUPLICATE DETECTION EVALUATION

**Setup** We benchmark RETSim's ability to identify near-duplicate content on real-world datasets from the literature. The NEWS-COPY Deduplication dataset (Silcock et al., 2022) contains 27,210 historical news articles with 122,876 positive duplicate pairs. The dataset consists of noisy near-duplicates due to factors like OCR errors, plagiarism, and news aggregation. We also evaluate the algorithms on the CORE Near-Duplicates dataset (Gyawali et al., 2020), which consists of 100k scholarly articles (title + abstract) with 25k exact duplicates, 25k near-duplicates, and 50k non-duplicates. Near-duplicates in this dataset arise from article revisions, versioning and metadata differences, and human typos. A key difference between these two benchmarks and the W4NT3D benchmark is that these two benchmarks are focused on detecting and clustering near-duplicate text, rather than robust text retrieval based on syntactic similarity. For both benchmarks, we follow the experimental setup provided in the papers and report Adjusted Rand Index (ARI) for the NEWS-COPY dataset and report precision/recall/F1 scores on the CORE Near-Duplicates dataset.

**Results** On the NEWS-COPY dataset, RETSim$_{\text{Partial-Dup}}$ outperforms all other approaches by a significant margin (4.8% ARI compared to our best MinHash result), as reported in Table 3. In the dataset, there are many near-duplicate pairs where one text is significantly longer than the other, so it is expected that RETSim$_{\text{Partial-Dup}}$, which can find matching text chunks in documents, is more suited for the task and outperforms RETSim$_{\text{Near-Dup}}$. Bucketing the near-duplicate detection rate of each algorithm by the length ratio between positive pairs (Figure 5), we observe that RETSim$_{\text{Partial-Dup}}$ outperforms MinHash regardless of the length ratio, but MinHash surpasses RETSim$_{\text{Near-Dup}}$ performance when one text is above roughly 1.5x the length of the other text in a near-duplicate pair.

| Model/Algorithm | ARI |
|---|---|
| Multilingual USE | 0.730 |
| Multilingual E5-Base | 0.742 |
| S-BERT* | 0.700 |
| SimHash | 0.695 |
| MinHash* | 0.737 |
| MinHash (Ours) | 0.783 |
| RETSim$_{\text{Partial-Dup}}$ | **0.831** |
| RETSim$_{\text{Near-Dup}}$ | 0.704 |

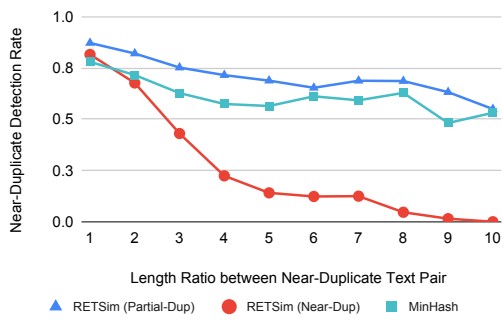

Table 3: Performance comparison on the NEWS-COPY dataset. Adjusted Rand Index (ARI) values are reported. * denotes results from Silcock et al. (2022).

Figure 5: Near-duplicate detection rate of RET-Sim vs MinHash for different length ratios of positive pairs. X-axis is the length of longer divided by shorter text, rounded to the nearest integer.

Additionally, we noticed that the labels in the dataset were occasionally noisy, as a substantial portion of the RETSim false positives appear to be near-duplicates upon inspection (Appendix A.6).

On the CORE Near-Duplicates dataset (Table 4), where documents are roughly the same size, RETSim$_{\text{Partial-Dup}}$ and RETSim$_{\text{Near-Dup}}$ performance is nearly the same. RETSim$_{\text{Partial-Dup}}$ outperforms MinHash and semantic text embedding baselines on 5 out of 6 reported metrics, including macro F1 score. We use MinHash + LSH with 256 hash functions for computational efficiency and for better performance than the default setting, as recommended by the datasketch library[1]. Deduplication thresholds and hyperparameter settings for the algorithms can be found in Appendix A.3.

| Model / Algorithm | Precision Duplicates | Recall Duplicates | Precision Non-Duplicates | Recall Non-Duplicates | Macro F1 | Accuracy |
|---|---|---|---|---|---|---|
| Exact Title Matching* | 0.830 | 0.500 | 0.709 | **0.992** | 0.757 | 0.746 |
| LaBSE | 0.937 | 0.923 | 0.930 | 0.943 | 0.933 | 0.919 |
| Multilingual USE | 0.917 | 0.907 | 0.918 | 0.927 | 0.917 | 0.909 |
| Multilingual E5-Base | 0.937 | 0.940 | 0.945 | 0.942 | 0.941 | **0.932** |
| MinHash + LSH | 0.929 | 0.902 | 0.915 | 0.938 | 0.921 | 0.918 |
| RETSim$_{\text{Partial-Dup}}$ | **0.945** | **0.941** | **0.945** | 0.949 | **0.945** | 0.928 |
| RETSim$_{\text{Near-Dup}}$ | 0.928 | 0.937 | 0.942 | 0.934 | 0.935 | 0.926 |

Table 4: Evaluation results on the CORE Near-Duplicates dataset. Precision/recall/macro F1 and accuracy numbers are reported. * denotes results from Gyawali et al. (2020).

## 5 APPLICATIONS

### 5.1 TRAINING DATASET DEDUPLICATION

| Model/Algorithm | % train examples with dup in train | % valid examples with dup in train |
|---|---|---|
| MinHash + LSH | 0.47% | 0.46% |
| Exact Substring* | 2.76% | 0.52% |
| RETSim$_{\text{Near-Dup}}$ | 3.17% | 0.59% |
| RETSim$_{\text{Partial-Dup}}$ | 12.77% | 2.66% |

Table 5: Deduplication rate on Wiki-40B (English). * denotes results from Lee et al. (2022).

**Setup** We evaluate RETSim's ability to deduplicate text training datasets by deduplicating the English split of Wiki-40B (Guo et al., 2020). We conservatively set the cosine similarity deduplication threshold to 0.1 for RETSim$_{\text{Near-Dup}}$ and 0.15 for RETSim$_{\text{Partial-Dup}}$ to limit the amount of false positives, based on the optimal thresholds found in the evaluation (Appendix A.3). We use USearch's default vector index for approximate nearest neighbor search (Vardanian, 2023). We compare

---

[1]datasketch: Big Data Looks Small. https://github.com/ekzhu/datasketch.

| Model/Algorithm | Accelerator | Batch Size | Embedding / Hashing time (sec) | Embedding / Hashing examples/sec |
|---|---|---|---|---|
| MinHash + LSH | CPU AMD 7950 32 cores | - | 234 | 12544 |
| RETSim | Onnx CPU AMD 7950 32 cores | 256 | 10839 | 270 |
| RETSim | TensorFlow GPU RTX 4090 | 4096 | 720 | 4062 |
| RETSim | TensorFlow GPU NVIDIA H100 | 16384 | 363 | 8069 |

Table 6: Embedding/hashing speed of RETSim vs MinHash + LSH on the Wiki-40B dataset.

against MinHash + LSH, where we set the number of hash functions to be 256 following Kocetkov et al. (2022) and use a Jaccard similarity threshold of 0.8 for deduplication (Lee et al., 2022).

**Results** Overall, as reported in Table 5, RETSim$_{\text{Near-Dup}}$ finds slightly more duplicates in the Wiki-40B training and validation splits. This is in-line with our deduplication results (Section 4.3) where RETSim$_{\text{Near-Dup}}$ outperforms other algorithms. On the other hand, RETSim$_{\text{Partial-Dup}}$ finds significantly more matches than the exact substring matching algorithm used in the previous study (Lee et al., 2022), showcasing the usefulness of performing both near-duplicate and partial-duplicate matching at once. This larger-than-expected number of partial matches indicate that machine learning practitioners should take extra care to deduplicate Wikipedia at the chunk level to avoid feeding duplicate text to their models.

In terms of embedding speed (Table 6), RETSim is significantly slower than MinHash + LSH on CPU (46x slower), competitive when using a desktop GPU such as the RTX 4090 (3x slower) and almost on-par when using a high-end GPU like the NVIDIA H100 (1.5x slower). Our current code is written in Python and not fully optimized, so we expect this performance gap to significantly shrink as we optimize our implementation. Although RETSim is slower than MinHash, RETSim is significantly smaller and faster than other text embedding models, and closes the performance gap between neural and non-neural based methods for near-duplicate text detection and dataset deduplication. Both RETSim$_{\text{Near-Dup}}$ and RETSim$_{\text{Partial-Dup}}$ are returned at the same time so they have the same embedding speed. Indexing and retrieval times will depend on the vector index and search algorithm used. For longer documents, RETSim$_{\text{Partial-Dup}}$ will produce more embeddings than RETSim$_{\text{Near-Dup}}$, so RETSim$_{\text{Partial-Dup}}$ offers a tradeoff between finer-grained matching versus indexing/retrieval speed, which will depend on the specific vector search algorithm and dataset used.

## 5.2 In the Wild: Spam Email Clustering

In this section, we showcase RETSim's real-world performance on clustering near-duplicate text which has been heavily manipulated by adversarial attacks by performing an evaluation on spam campaigns. Spam constitutes a strong proving ground for near-duplicate clustering algorithms as spammers employ adversarial augmentation techniques in an attempt to evade detection. Such augmentations typically include appending or prepending unrelated text, interleaving random words and different languages, intentionally introducing typos, abusing extended character sets such as emojis and homoglyphs, and more. These techniques are collectively referred to as hash-busting.

**Setup** The dataset consists of 5,252 spam emails from 196 spam campaigns, donated by Gmail users who flagged them when they reached their inboxes. Each example contains the email subject concatenated with the message content. The emails were misclassified by a spam classifier due to their effective adversarial text manipulation techniques, which makes them a challenging test set for clustering evaluations. Some examples of hash-busting attacks and adversarial manipulations we observe include the use of homoglpyphs, uncommon Unicode character sets, invisible characters, and padding with random words from different languages. To get the ground truth campaign clusters, emails were manually reviewed and assigned to a specific spam campaign based on similarity by human reviewers. We use agglomerative clustering to cluster spam emails, and report homogeneity, completeness, V-Measure, and Adjusted Rand Index (ARI) metrics.

**Results** Overall, we observed that RETSim is significantly better at clustering near-duplicates with adversarial manipulations, outperforming both SimHash and USE across all metrics considered (Table 7). In particular, we observed that RETSim outperforms USE by 4.6% on the V-Measure score which is our main metric. The results reported in this section are in-line with what we observe since we deployed RETSim as our main near-duplicate detection algorithm in December 2022.

| Model / Algorithm | Homogeneity | Completeness | V-Measure | ARI |
|---|---|---|---|---|
| USE | 0.856 | 0.955 | 0.903 | 0.6 |
| SimHash + LSH | 0.867 | 0.876 | 0.871 | 0.571 |
| RETSim$_{\text{Near-Dup}}$ | **0.937** | **0.963** | **0.949** | **0.747** |

Table 7: Performance on clustering adversarial spam campaigns in practice.

# 6 ABLATION STUDIES

**Setup** In this section, we summarize the key ablation studies we performed when designing RET-Sim. All the models used in this section are trained using the setup detailed in Appendix A.1.2, except we only train them for 100k steps to reduce computational costs. We evaluate RETSim$_{\text{Near-Dup}}$'s performance for each model on a subset of the W4NT3D benchmark, where we randomly select 1000 examples from each of the 41 language splits and use Recall@1 as reported metric.

| Block Type | Recall@1 | | Chunk Size | Recall@1 | | Embed. Dim | Recall@1 |
|---|---|---|---|---|---|---|---|
| RETVec MLP | 0.975 | | 128 | 0.979 | | 64 | 0.969 |
| ConvNeXt | 0.978 | | 256 | 0.984 | | 128 | 0.980 |
| BERT | 0.973 | | *512 | 0.986 | | *256 | 0.986 |
| T5 | 0.980 | | 1024 | 0.983 | | 512 | 0.986 |
| *GAU | 0.986 | | 2048 | 0.978 | | 768 | 0.986 |

Table 8: RETSim ablation study results on architecture block type (left), text chunk size (middle), and embedding dimension (right). ***Bold*** denotes the value selected for the final RETSim model.

**Results** Table 8 contains RETSim ablation study results on max text chunk size, architecture block type, and embedding size. The most important architectural decision was to decide the optimal text chunk size and finding the right balance between having the smallest size possible to maximize RETSim$_{\text{Partial-Dup}}$ efficiency while ensuring RETSim$_{\text{Near-Dup}}$ full-text embeddings can work effectively on full documents. We find that chunks of 512 characters offer the best performance.

We also tested various model architectures and transformer blocks to find the best balance between efficiency and performance. We find that the more modern GAU block (Hua et al., 2022) outperforms the vanilla BERT transformer block (Devlin et al., 2019) and the T5 block (Xue et al., 2020). We also tried modern CNN architectures such as ConvNeXt (Liu et al., 2022) and the MLP architecture proposed in RETVec (Bursztein et al., 2023), but both were worse than GAU block performance. Last but not least, we found that increasing the embedding size past 256 dimensions does not yield any meaningful improvements for RETSim$_{\text{Near-Dup}}$. Accordingly, we opted to use a 256-dimension embedding for space-efficiency and to maximize indexing and query speed. Additional ablation studies for other hyperparameters can be found in Appendix A.5.

# 7 FUTURE WORK

RETSim's novel training regime, which combines metric learning and data augmentation, has many other potential applications that we plan to explore in future work. For example, it could be adapted or extended to train robust semantic embeddings or image similarity embeddings. Additionally, we expect that as general models become bigger and more expensive to run in the future, smaller, specialized models such as RETSim will emerge as an efficient alternative for a wide range of tasks.

# 8 CONCLUSION

In this paper, we introduced RETSim, a novel, multilingual text embedding which achieves state-of-the-art performance on near-duplicate text detection, dataset deduplication, and syntactic text similarity benchmarks. RETSim is significantly faster than previous neural-based text embeddings and more robust than n-gram based algorithms, which makes it suitable for large-scale text retrieval and dataset deduplication, especially in adversarial settings such as spam detection. Furthermore, we introduced the W4NT3D benchmark, the first multilingual dataset designed to measure the adversarial robustness of near-duplicate text detection algorithms. We release both RETSim and the W4NT3D benchmark under the MIT License.

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

## A APPENDIX

### A.1 RETSIM DETAILS

#### A.1.1 RETSIM MODEL HYPERPARAMETERS

The full list of RETSim model hyperparameters can be found in Table 9.

| Hyperparameter | Value |
|---|---|
| Max input length (per chunk) | 512 |
| Block type | GAU |
| # blocks | 2 |
| Hidden dim | 256 |
| Expansion rate | 1 |
| Activation function | Swish |
| Attention activation function | $relu^2$ |
| Absolute positional encoding | ScaledSin |
| Relative positional encoding | RoPE |
| Norm type | ScaleNorm |
| Pooling type | GeM $(p = 3)$ |
| Dropout rate | 0 |
| Embedding dim | 256 |
| # Parameters | 536k |

Table 9: Detailed RETSim model hyperparameters.

#### A.1.2 RETSIM TRAINING HYPERPARAMETERS

Table 10 details the hyperparameters settings for training configuration, loss, and optimizer used to train the RETSim model.

| Hyperparameter | Value |
|---|---|
| Batch size | 1024 |
| Train steps | 1 million |
| LAMB $\epsilon$ | 1e-6 |
| LAMB $\beta_1$ | 0.9 |
| LAMB $\beta_2$ | 0.999 |
| Max learning rate | 0.001 |
| End learning rate | 0 |
| Learning rate decay | Cosine |
| Weight decay | 0 |

Table 10: RETSim detailed training hyperparameters.

### A.2 TRAINING DATASET DETAILS

Below, we provide the full list of augmentations used to generate augmented text for the RETSim training dataset, as described in Section 3.2.

SENTENCE-LEVEL AUGMENTATIONS

- Deletion:
  - Random sentence deletion
  - Random sentence truncation
- Insertion:

  - – Random prefix sentence
  - – Random suffix sentence
  - – Random sentence insertion
  - – Repeat sentence
- Substitution:
  - – Lowercase/uppercase sentence
  - – Random sentence substitution
- Transposition:
  - – Neighboring Swap

## WORD-LEVEL AUGMENTATIONS

- Deletion:
  - – Random word deletion
- Insertion:
  - – Random word insertion
  - – Random word insertion per language
- Substitution:
  - – 3-gram frequency based word substitution
  - – Random word substitution
  - – Random word substitution per language
  - – Repeat word
- Transposition:
  - – Neighboring Swap

## CHARACTER-LEVEL AUGMENTATIONS

- Deletion:
  - – Random character deletion
- Substitution:
  - – Case substitution
  - – n-gram based substitution for $n = 3, 4, 5$
  - – QWERTY keyboard typo substitution
  - – Homoglyphs substitution
  - – Random ASCII substitution
  - – Random character from language alphabet substitution
  - – Random punctuation substitution
  - – Random Unicode character substitution
- Insertion:
  - – Character repetition
  - – n-grams based insertion for $n = 3, 4, 5$
  - – Random character from language alphabet insertion
  - – Random punctuation insertion
  - – Random Unicode character insertion
- Transposition:
  - – Neighboring swap

### A.3 DETAILED EVALUATION HYPERPARAMETERS

Figures 6 and 7 contain information on deduplication thresholds values and hyperparameter settings for each algorithm benchmarked on the NEWS-COPY and CORE deduplication datasets.

| Model / Algorithm | Threshold Type | Threshold Value | Hyperparameters |
|---|---|---|---|
| Multilingual USE | Cosine Similarity | 0.88 | - |
| Multilingual E5-Base | Cosine Similarity | 0.96 | - |
| SimHash | Hamming Distance | 10 | 64 bits, 5-grams (character-level) |
| MinHash (Ours) | Jaccard Similarity | 0.6 | 10 hash functions, 2-grams (word-level) |
| RETSim$_{Near-Dup}$ | Cosine Similarity | 0.89 | - |
| RETSim$_{Partial-Dup}$ | Cosine Similarity | 0.84 | - |

Figure 6: Hyperparameter settings for NEWS-COPY dataset evaluation in Section 4.3.

| Model / Algorithm | Threshold Type | Threshold Value | Hyperparameters |
|---|---|---|---|
| LaBSE | Cosine Similarity | 0.88 | - |
| Multilingual USE | Cosine Similarity | 0.87 | - |
| Multilingual E5-Base | Cosine Similarity | 0.96 | - |
| SimHash + LSH | Hamming Distance | 6 | 64 bits, 3-grams (character-level) |
| MinHash + LSH | Jaccard Similarity | 0.5 | 256 hash functions, 3-grams (word-level) |
| RETSim$_{Near-Dup}$ | Cosine Similarity | 0.86 | - |
| RETSim$_{Partial-Dup}$ | Cosine Similarity | 0.82 | - |

Figure 7: Hyperparameter settings for CORE Near-Duplicates dataset evaluation in Section 4.3.

### A.3.1 DEDUPLICATION THRESHOLD IMPACT

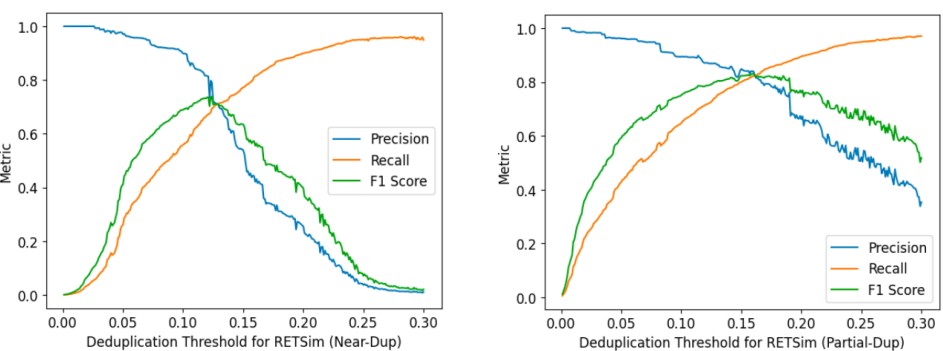

Figure 8: Precision/Recall/F1 scores for different cosine distance deduplication thresholds for RETSim$_{Near-Dup}$ (left) and RETSim$_{Partial-Dup}$ (right) on the NEWS-COPY dataset.

### A.4 DETAILED W4NT3D BENCHMARK RESULTS

Tables 11 and 12 show detailed performance results for RETSim and all baseline algorithms for every language split in the W4NT3D benchmark.

| Model / Algorithm | ar | bg | ca | cs | da | de | el | en | es | et | fa | fi | fr | he | hi | hr | hu | id | it | ja | ko |
|---|---|---|---|---|---|---|---|---|---|---|---|---|---|---|---|---|---|---|---|---|---|
| LaBSE | 0.915 | 0.915 | 0.897 | 0.912 | 0.938 | 0.931 | 0.918 | 0.944 | 0.888 | 0.923 | 0.912 | 0.926 | 0.930 | 0.937 | 0.927 | 0.912 | 0.898 | 0.915 | 0.929 | 0.931 | 0.949 |
| Multilingual USE | 0.915 | 0.909 | 0.883 | 0.900 | 0.927 | 0.942 | 0.874 | 0.958 | 0.903 | 0.930 | 0.870 | 0.913 | 0.938 | 0.841 | 0.648 | 0.889 | 0.885 | 0.928 | 0.937 | 0.990 | 0.984 |
| Multilingual E5-Base | 0.936 | 0.899 | 0.890 | 0.927 | 0.956 | 0.944 | 0.934 | 0.959 | 0.896 | 0.951 | 0.929 | 0.943 | 0.948 | 0.958 | 0.964 | 0.901 | 0.889 | 0.942 | 0.933 | 0.979 | 0.986 |
| PaLM 2 (Gecko) | 0.497 | 0.851 | 0.902 | 0.911 | 0.943 | 0.932 | 0.589 | 0.961 | 0.911 | 0.926 | 0.356 | 0.914 | 0.934 | 0.589 | 0.435 | 0.903 | 0.876 | 0.924 | 0.937 | 0.578 | 0.701 |
| SimHash | 0.558 | 0.510 | 0.485 | 0.579 | 0.592 | 0.561 | 0.557 | 0.591 | 0.496 | 0.665 | 0.568 | 0.632 | 0.519 | 0.651 | 0.563 | 0.557 | 0.530 | 0.533 | 0.513 | 0.465 | 0.593 |
| MinHash | 0.633 | 0.506 | 0.508 | 0.581 | 0.598 | 0.558 | 0.574 | 0.591 | 0.504 | 0.585 | 0.595 | 0.568 | 0.556 | 0.697 | 0.606 | 0.556 | 0.512 | 0.563 | 0.575 | 0.223 | 0.814 |
| RETSim$_{Partial-Dup}$ | 0.928 | 0.942 | 0.924 | 0.950 | 0.953 | 0.949 | 0.935 | 0.954 | 0.938 | 0.953 | 0.946 | 0.956 | 0.947 | 0.945 | 0.941 | 0.934 | 0.950 | 0.950 | 0.944 | 0.963 | 0.971 |
| RETSim$_{Near-Dup}$ | 0.971 | 0.976 | 0.968 | 0.973 | 0.988 | 0.978 | 0.974 | 0.987 | 0.976 | 0.985 | 0.971 | 0.981 | 0.983 | 0.989 | 0.975 | 0.962 | 0.962 | 0.976 | 0.982 | 0.986 | 0.991 |

Table 11: Full per-language Recall@1 performance for various embedding models and algorithms on the W4NT3D benchmark (part 1).

| Model / Algorithm | lt | lv | ms | nl | no | pl | pt | ro | ru | sk | sl | sr | sv | th | tl | tr | uk | vi | zh-cn | zh-tw |
|---|---|---|---|---|---|---|---|---|---|---|---|---|---|---|---|---|---|---|---|---|
| LaBSE | 0.919 | 0.922 | 0.919 | 0.931 | 0.928 | 0.928 | 0.944 | 0.909 | 0.918 | 0.922 | 0.931 | 0.930 | 0.906 | 0.882 | 0.947 | 0.930 | 0.899 | 0.932 | 0.917 | 0.918 |
| Multilingual USE | 0.902 | 0.919 | 0.932 | 0.936 | 0.921 | 0.931 | 0.952 | 0.869 | 0.910 | 0.901 | 0.908 | 0.906 | 0.899 | 0.888 | 0.949 | 0.940 | 0.893 | 0.910 | 0.986 | 0.985 |
| Multilingual E5-Base | 0.941 | 0.935 | 0.949 | 0.934 | 0.944 | 0.928 | 0.955 | 0.936 | 0.911 | 0.921 | 0.940 | 0.896 | 0.925 | 0.921 | 0.969 | 0.951 | 0.872 | 0.945 | 0.980 | 0.978 |
| PaLM 2 (Gecko) | 0.909 | 0.907 | 0.928 | 0.931 | 0.930 | 0.913 | 0.950 | 0.893 | 0.851 | 0.924 | 0.916 | 0.856 | 0.919 | 0.571 | 0.944 | 0.902 | 0.822 | 0.863 | 0.623 | 0.609 |
| SimHash | 0.609 | 0.624 | 0.533 | 0.527 | 0.577 | 0.586 | 0.548 | 0.514 | 0.554 | 0.575 | 0.580 | 0.553 | 0.552 | 0.669 | 0.507 | 0.606 | 0.517 | 0.609 | 0.276 | 0.315 |
| MinHash | 0.568 | 0.579 | 0.511 | 0.540 | 0.560 | 0.563 | 0.573 | 0.520 | 0.523 | 0.573 | 0.570 | 0.553 | 0.520 | 0.416 | 0.570 | 0.583 | 0.520 | 0.581 | 0.172 | 0.200 |
| RETSim$_{Partial-Dup}$ | 0.957 | 0.945 | 0.961 | 0.950 | 0.958 | 0.953 | 0.954 | 0.948 | 0.946 | 0.961 | 0.947 | 0.963 | 0.953 | 0.941 | 0.961 | 0.954 | 0.941 | 0.951 | 0.946 | 0.957 |
| RETSim$_{Near-Dup}$ | 0.980 | 0.983 | 0.980 | 0.979 | 0.981 | 0.979 | 0.985 | 0.977 | 0.970 | 0.971 | 0.977 | 0.979 | 0.969 | 0.946 | 0.989 | 0.978 | 0.957 | 0.985 | 0.971 | 0.968 |

Table 12: Full per-language Recall@1 performance for various embedding models and algorithms on the W4NT3D benchmark (part 2).

## A.5 Additional Ablation Studies

This section includes ablation studies on additional hyperparameters for the RETSim model, including the loss function, pooling type, and model capacity.

| $\alpha$ | $\beta$ | $\lambda$ | Recall@1 |
|---|---|---|---|
| 2 | 20 | 0.5 | 0.982 |
| 2 | 20 | 1 | 0.948 |
| 2 | 40 | 0.5 | 0.984 |
| 2 | 40 | 1 | 0.919 |
| 4 | 20 | 0.5 | 0.982 |
| 4 | 20 | 1 | 0.947 |
| **4** | **40** | **0.5** | **0.986** |
| 4 | 40 | 1 | 0.923 |

Table 13: Ablation study on Multi-Similarity Loss hyperparameters for RETSim training. **Bold** indicates the hyperparameter setting selected for the final model.

| # Blocks | Hidden Dim | Recall@1 |
|---|---|---|
| 2 | 64 | 0.965 |
| 2 | 128 | 0.980 |
| **2** | **256** | **0.986** |
| 2 | 512 | 0.986 |
| 3 | 64 | 0.962 |
| 3 | 128 | 0.980 |
| 3 | 256 | 0.984 |
| 3 | 512 | 0.987 |
| 4 | 64 | 0.966 |
| 4 | 128 | 0.980 |
| 4 | 256 | 0.985 |
| 4 | 512 | 0.986 |

Table 14: Ablation study for RETSim model capacity and size (number of GAU blocks and hidden dimension for the blocks). **Bold** indicates the hyperparameter setting selected for the final model.

| Pooling Type | Recall@1 |
|---|---|
| Average Pooling | 0.985 |
| Max Pooling | 0.983 |
| **Generalized Mean Pooling** | **0.986** |

Table 15: Ablation study on pooling type for the RETSim model. **Bold** indicates the hyperparameter setting selected for the final model.

## A.6 Selected Examples from NEWS-COPY Dataset

In this section, we randomly selected a set of false positives and false negatives for RETSim on the NEWS-COPY deduplication dataset to provide further insight into the results.

| Text 1 | Text 2 |
|---|---|
| chauffeur, a policeman and a passing journalist who tried to intervene. Beaton and the policeman were reported in serious condition. The 23-year-old princess and her husband of five months, Capt. Mark Phillips, were not hurt. But police experts said the holes left by one of the bullets fired into the car indicated it passed between them, missing them by inches. A police informant said it was believed 11 shots were fired by the assailant. Experts were studying two revolvers found at the scene. They said fi... | 'LONDON (AP) — Ian Ball, a 26-year- old unemployed Englishman, was brought into court today and charged with attempted murder during an at- tempt to kidnap Princess Anne from her car in the heart of London Wed- nesday night. Ball, lean-faced and bearded, stood stiffly in the dock at the Bow Street Magistrate's court, handcuffed to two detectives. He spoke only once during his 60-second appearance, saying iha London accent: "I want to apply for legal aid." The court or- dered him held for another hearing on Ma... |
| By United Press tnfernational Ay SSAST OR BE FRE NG SG The federal government has proposed new methods of eoustructing federal buildings in a move to save ad- ditional energy and suggested ils elfort could be adapted to all new buildings, | Hy United Press International The federal government has Proposed new methods of constructing federal buildings in a move lo save addilional energy and suggested ils effort could be adapted to all new buildings, Arthur F, Sampson, General Services Administration ad- ministrater, said new features for such construction would include the collection of rain waler for cooling and irriga- tion, solar energy collectors and the covering of exterior walls with earth. "Whal we are saying is that these design criteri... |
| Washington, Jan. 27. —(P)—Im- mediate removal of John F. J. Her- bert, as prohibition administrator for Montana and Idaho, was de- manded in the senate today by Sen- ators Borah, Idaho, and Wheéeler, Montana, on the ground of charges placed before them by department of justice investigators. Wheeler accompanied his demand (Continued on Page 2) | — Washington, Jan. 27 1 AP).—Immiedl- aie mmoval of John F. Herbert as pro- — hi- bition administrator for Montana and 'Idaho was demanded m the Seuate to- 'day by Sen- ators Borah. idaho, and Waeeler, Montana. on the ground of charges placed before them by Depart- meat of Justice investigators. Wheeler accompanied his demand nith a declaration that prohibition en- foreemen: had brukea down. He blamed the "politicians" and called upon the Law Enforcement Commussion to sum- mon members of the Republican Na- tona... |
| By RAYMOND CLAPPEA (Dnited Presa Stal Correspandoayy London, Jai, 38—(UP—-The Am 'erlcnn delegation to the navat confer ence today won ls demand for pre- sentation: of the cnse of suxiliary warships limitation first at tho next plenary session Thuvaday, 'Tho chlet delegates, mec- tittg at St. James palace, also decided that tho plenary sesslon would discuss the Main con- ference questions in alpha betical order of ihe countriea pro- posing. Press ta be Admitted The American delegation woo a second vic- tory whe... | London, Jan. 24, W.P—The Amer- jean dele- gation fo the naval cen- ference teday won its demand for presentation of the case of auxil- jary warships linsitation flrst at the next ple- trary session 'Vhursday, The chief delegates, meeting at Si, James Pelace, also decided that the plenary session would discuss the main confeyence questions in alphabetical order af the cauntries proposing. The American del- egation won a second victory when it was decided to udmil certain representatives of the press at fie plenary... |

Table 16: Example false negatives for RETSim on the NEWS-COPY dataset (pairs of texts not detected as near-duplicates by RETSim but labeled as near-duplicates in the original dataset). Examples are randomly selected and truncated at 512 characters for display.

| Text 1 | Text 2 |
|---|---|
| BOZEMAN, Mont. (AP) — Chet Huntley, whose resonant voice and rough-hewn face be- came familiar to millions on the nightly television news, died Wednesday in his mountain resort home. He was 62. He underwent surgery for lung cancer in January but had remained activesuntil recent weeks. He died at 2:20 a.m, according to his widow, Tippy Hunt.cy. Huntiey was teamed for 14 years with David Brinkley on NBC's Huntley-Brinkley Re- port. He quit in 1970 and re-turned to his native Montana to develop the $20-millio... | BOZEMAN, Mont. (AP) - Chet Huntley, whose resonant voice and rough-hewn face became familiar to millions on the nightly television news, died Wednesday in his mountain resort home. He was 62. He underwent surgery for lung cancer in January but had remained active until recent weeks. He died at 2:20 a.m., according to his widow, Tippy Huntley. Huntley was teamed for 14 years with David Brinkley on NBC's Huntley-Brinkley Report. He quit in 1970 and returned to his native Montana to develop the $20 mil-lion Bi... |
| By THE ASSOCIATED PRESS Some Amer-icans are paying up to 50 per cent more per month for electricity this year than they did last, an Associ- ated Press survey shows. Con-sumers are beginning to organize to fight the rate hikes. A spot check of monthly elec- tric bills this year and last showed that most in-creases have been about $1 or $2, gen- erally about 10 per cent, with the highest reported boost com- ing in Jacksonville, Fia., where the average tab went from $17.90 last year to $27.70 this year. Utility... | By Louise Cook Acenciaiod Prece Writer Same Americans are paying up io 20 per cent more per month far electricity this year ihan they did last, an -Associ- Press survey shows. onsumers are beginning to ze to fight the rate hikes, A spot check of monthly elec- tre hills this year and Jast showed that most increases ve been about $1 or $2, gen- erally about 10 per cent, with the highest reported boost com-ing in Jacksonville, Fla., where the average tab went from $17.90 last year to $27.70 this year... |
| BOZEMAN, Mont. (AP) — Vice President Gerald R. Ford says the world will miss the "'unique abilities" of former television news anchorman Chet Huntley. Huntley, 62, died at his home Wednesday after a long bout with lung cancer. Family 'spokesmen said a memorial service would be conducted for Huntley Sunday at the Big Sky of Montana' resort and recreation area south of Bozeman. Huntley was chairman of the Big Sky board of directors. Another memorial service is scheduled Tuesday in the New York studios of the... | BOZEMAN, Mont. (AP) — Vice President Gerald R. Ford says the world will miss the "unique abilities" of former television news anchorman Chet Huntley. Huntley, 62, died at his home Wednesday after a long bout with lung cancer. Family spokesmen said a me-morial service would be con- ducted for Hunt-ley Sunday at the Big- Sky of Montana resort and recreation area south of Bozeman. Hunt-ley was chair- man of the Big Sky board of directors. Another memorial service is sched-uled Tuesday in the New York studios of... |
| WASHINGTON (AP) — The House has passed legislation raising the minimum wage from $1.60 an hour to $2 this year for most workers covered and to $2.30 for all by 1978. The bill, approved Wednesday 375 to 37, also would increase by 7 million to 56.5 million the number of workers covered by the mini-mum wage laws. The bill is a modified ver-sion of one President Nixon vetoed last year. However, he is expected to sign this one if it is finally approved after ad- justment with a similar Senate passed measure, altho... | ˍ WASHINGTON (AP) — The House has passed legislation raising the minimum wage from $1.60 an hour to $2 this year for most workers covered and to $2.30 for all by 1978. The bill, approved Wednes- day 375 to 37, also would in- crease by 7 million to 56.5 mil-lion the number of workers cov- ered by the minimum wage laws. The bill is a modified version of one President Nixon vetoed last year. However, he is ex- ted to sign this one if it is inally approved after adjust- ment with a similar Senate- passed measu... |

Table 17: Example false positives for RETSim on the NEWS-COPY dataset (pairs of texts detected as near-duplicates by RETSim but not labeled as near-duplicates in the original dataset). Examples are randomly selected and truncated at 512 characters for display.

