# OpenReview forum: "RETSim: Resilient and Efficient Text Similarity"
_ICLR.cc/2024/Conference — ICLR 2024 poster_

### Official Review · Reviewer_XEvv · 2023-10-31

**Soundness:** 2 fair
**Presentation:** 2 fair
**Contribution:** 1 poor
**Rating:** 6
**Confidence:** 4

**Summary:**

the authors build a more light-weighted embedding for document similarity detection task, and conducted extensive experiment to prove the performance of their design. On their purposed dataset, they prove that the new model/embedding has better overall performance in multilingual setting retrieval. Overall the paper is complete and easy to follow. However there is not much original contribution in terms of model design, and the improvement seems incremental and marginal. See comments below

**Strengths:**

extensive experimentation - from the purposed multilingual dataset W4NT3D to NEWS-COPY dataset, the experiment is thorough and convincing in demonstrating purposed effectiveness
efficiency over other nn models - personally I feel the paper is most valuable in that the purposed model is much simpler than the rest deep models, while their embedding maintains a relatively same or even better performance, although it's on the purposed dataset. the saving from efficiency instead of performance is the essence

**Weaknesses:**

originality - the purposed model arch is more like a concatenation of existing methods/building blocks, and frankly speaking I can see very little original or significant contribution
incremental contribution - the improvement is claimed on the purposed multilingual dataset, while it's not substantially surpassing the other deep model counterparties. for the real world duplication detection task, the traditional minhash based method is a better choice with decent performance IMHO, considering computation constraint

**Questions:**

table 4 column 4 - should be recall non-duplicates?
palm 2 gecko embedding performed much worse in the paper than it should be. any hypothesis or analysis on this?

---

> ### Author Response · Authors · 2023-11-16
> **Response to Reviewer XEvv (1/2)**
>
> Thank you for the review and constructive feedback. We appreciate that you thought the paper’s experiments were extensive and convincing, as well as the value in the proposed method being more performant than other deep learning-based methods. Additionally, we have revised the paper based on feedback from the reviewers to expand upon our methodology, highlight the novelty of our proposed approach, and added additional experiments.
>
> Please find our responses to your comments and questions below.
>
> > originality - the purposed model arch is more like a concatenation of existing methods/building blocks, and frankly speaking I can see very little original or significant contribution incremental contribution
>
> Indeed, RETSim builds upon existing techniques and model architectures to achieve state-of-the-art performance across the benchmarks. However, we kindly disagree that there is little novelty in our approach. To the best of our knowledge, using metric learning to train robust embeddings for full text documents is a novel approach that has not been explored in prior work. Furthermore, our architecture which outputs both partial and global embeddings in a single, efficient forward pass is a novel refinement unseen in prior neural-based text embeddings. While these advances may seem small, they are fundamental to helping RETSim outperform MinHash (the dominant algorithm for over 20 years) on a large variety of real-world, near-duplicate text detection and clustering experiments, including standard benchmarks from prior literature. Last but not least, we believe our new W4NT3D dataset can contribute to advancing the broader field by providing a new multilingual benchmark to measure the robustness of near-duplicate text retrieval algorithms.
>
> >  the improvement is claimed on the purposed multilingual dataset, while it's not substantially surpassing the other deep model counterparties.
>
> We agree that all models achieve high performance on the multilingual dataset, but the RETSim model is more than 100x smaller than other text embedding models, making RETSim far more scalable and efficient. RETSim also far suprasses MinHash and SimHash on the multilingual dataset in terms of performance.
>
> While we did create our own benchmark (W4NT3D) because of the lack of existing multilingual near-duplicate detection benchmarks, we took great care to reproduce popular benchmarks and datasets used in prior works. For all benchmarks, including the NEWS-COPY deduplication dataset and Wiki-40B, RETSim consistently outperforms other algorithms which supports our claim that it achieves new state-of-the-art performance. We would love to include any other relevant benchmarks that we may have missed if you have any suggestions as we strive to evaluate RETSim in the most comprehensive way possible.
>
> > for the real world duplication detection task, the traditional minhash based method is a better choice with decent performance IMHO, considering computation constraint
>
> We agree that MinHash is computationally efficient for large-scale dataset deduplication. However, we kindly disagree with the reviewer that MinHash is a better choice for real world deduplication tasks.
>
> MinHash is known to suffer from lack of resilience to typos and noise, which causes it to miss a significant fraction of near-duplicates. For example, on the real-world NEWS-COPY near-duplicate detection benchmark, RETSim outperforms MinHash by 4.8% ARI. For training corpus deduplication, RETSim is capable of finding many more near-duplicates than MinHash (over 5x on Wiki-40B English corpus). This is an important finding since it’s been shown that large language models regenerate sequences shown in the training dataset at a superlinear rate vs the sequence’s count in the training set, and that deduplicating training data helps secure a model against privacy attacks and reduce memorization [1, 2].
>
> Overall, RETSim offers a tradeoff between better performance (better near-duplicate detection, increased adversarial robustness) vs speed compared to MinHash. We strongly believe that there are many important use cases such as copyright detection, mitigating privacy risk, and security and anti-abuse applications where RETSim will be the best choice over MinHash. For example, as highlighted in Section 5.2, RETSim’s adversarial robustness is critical to improve clustering performance when dealing with email spam.
>
> [1] Kandpal et al. “Deduplicating Training Data Mitigates Privacy Risks in Language Models.” arxiv.org/abs/2202.06539, 2022.
>
> [2] Lee et al. “Deduplicating Training Data Makes Language Models Better.” arxiv.org/abs/2107.06499, 2021.

---

> ### Author Response · Authors · 2023-11-16
> **Response to Reviewer XEvv (2/2)**
>
> > Q1: table 4 column 4 - should be recall non-duplicates?
>
> Yes, thank you very much for catching that. It is fixed in the revised PDF.
>
> > Q2:  palm 2 gecko embedding performed much worse in the paper than it should be. any hypothesis or analysis on this?
>
> We are also unsure as to why PaLM 2 Gecko embeddings are not as performant as expected. PaLM 2 actually performs the best out of all baseline algorithms on the English subset of the W4NT3D benchmark (96.1% Recall@1), which makes us confident that our implementation is correct and it is likely tied to issues with multilingual performance on this benchmark. It is difficult to hypothesize why this is the case, since there are limited details regarding PaLM 2 Gecko embeddings available publicly.
>
> Please let us know if you have any additional questions or further suggestions, we are more than happy to revise and improve the paper. Thank you!

---

> > ### Comment · Reviewer_XEvv · 2023-11-22
> >
> > I acknowledge that I have read all the comments from authors and other reviewers. In general I feel the paper has its value though personally I would have a slightly higher bar for the topic. I decide to raise my score to marginally above.

---

### Official Review · Reviewer_rc5k · 2023-10-31

**Soundness:** 3 good
**Presentation:** 3 good
**Contribution:** 3 good
**Rating:** 8
**Confidence:** 4

**Summary:**

This paper proposes RETSim, a lightweight text embedding model designed for near-duplicate text retrieval, clustering, and dataset deduplication. RETSim is a cascade system comprising three models: a character-level vectorizer, a small transformer model, and an embedding averaging module. There are two variants, RETSim (Near-Dup) and RETSim (Partial-Dup), tailored for full matching and partial matching, respectively. The authors evaluate RETSim across multiple datasets, including their self-proposed W4NT3D benchmark, the two news deduplication dataset, and a span email clustering dataset, demonstrating its superiority over neural and hashing baselines.

**Strengths:**

- The proposed model is effective and efficient.
- RETSim trained on mC4 can be applied to a variety of datasets.

**Weaknesses:**

- The experiment in Section 5.1 makes no sense. Apparently, according to previous experiments, RETSim should be able to find more duplicates. How using RETSim may impact downstream language model pre-training remains unclear.
- Some baselines are missing in Table 4 and Table 7.
- Certain aspects of the method are not clearly explained, such as the process for obtaining the (512, 24)-dimension tensor from the vectorizer (Section 3.1) and the Multi-Similarity Loss, which appears to be a less common concept that deserves more detailed explanation (Section 3.2).
- The paper lacks a dedicated Conclusion section, with Section 7 not effectively serving as a substitute conclusion.

**Questions:**

- It seems that the dataset used in the experiment in Section 5.2 is a non-open dataset. The authors should provide more details on the dataset.
- For a same amount of input, does RETSim (Near-Dup) produce more embeddings than RETSim (Partial-Dup)? If yes, why isn’t the speed separately measured in Table 6? (I think RETSim (Partial-Dup) is slower, since it needs to retrieve against a significantly larger set of text embeddings during de-duplication?)

---

> ### Author Response · Authors · 2023-11-16
> **Response to Reviewer rc5k (1/2)**
>
> We thank the reviewer for their detailed review and helpful comments, which helped us improve the paper. Based on your suggestions, we have revised the paper to add three additional baselines to Table 4, added a separate conclusion section, and expanded upon our methodology to improve clarity.
>
> Please find our detailed responses to your questions below and the updated paper PDF for the revisions.
>
> > Q1: The experiment in Section 5.1 makes no sense. Apparently, according to previous experiments, RETSim should be able to find more duplicates. How using RETSim may impact downstream language model pre-training remains unclear.
>
> Deduplicating training corpuses is very important for large language models, as it is shown to improve performance as well as reduce memorization and privacy risks [1, 2]. In Section 5.1, we demonstrate that RETSim is capable of finding substantially more duplicates than MinHash, especially for partially-matching documents. We agree that empirically evaluating the impact of using RETSim vs MinHash to deduplicate training corpuses for large language models will be a great addition to the paper. We are looking into adding this experiment for the final revision to see if it produces meaningful improvements.
>
> [1] Kandpal et al. “Deduplicating Training Data Mitigates Privacy Risks in Language Models.” arxiv.org/abs/2202.06539, 2022.
>
> [2] Lee et al. “Deduplicating Training Data Makes Language Models Better.” arxiv.org/abs/2107.06499, 2021.
>
> > Q2: Some baselines are missing in Table 4 and Table 7.
>
> Thank you for the suggestions, we have revised Table 4 in the paper to add additional baseline results for neural-based text embeddings (Multilingual USE, Multilingual E5-Base, and LaBSE). RETSim outperforms these additional baselines for the main metrics as well (macro F1 and accuracy). Please see the updated paper for additional details.
>
> For Table 7, the spam email clustering experiment is run on production data, so we are only allowed to run vetted models. Thus, models like Multilingual E5-Base are not available for us to use, so we are unable to include it as a baseline. SimHash + LSH was the original algorithm being used in production so it was selected as a baseline. Due to data retention policies, we are unable to add additional baselines to this experiment at the moment but will look into rerunning the experiment for the final revision of the paper.
>
> > Q3: Certain aspects of the method are not clearly explained, such as the process for obtaining the (512, 24)-dimension tensor from the vectorizer (Section 3.1) and the Multi-Similarity Loss, which appears to be a less common concept that deserves more detailed explanation (Section 3.2).
>
> Thank you for the feedback, we agree and we have updated the PDF to include more detail and background context for the text vectorizer used in RETSim and for Multi-Similarity Loss. Please see Section 2 and Section 3 in the revised paper for the updates.
>
> We use the RETVec character encoder to obtain the (512, 24)-dimensional tensor representing the input text. The character encoder produces a binary representation of each Unicode character, which is then padded to 24 bits. This character-based representation has been shown to be compact, efficient, and resilient in prior work, which is why it was selected as the vectorizer for RETSim.
>
> We also agree that Multi-Similarity Loss is less known compared to alternatives such as Triplet Loss. Multi-Similarity Loss is based on a general weighting framework for pair-based losses and has been shown to be state-of-the-art for pair-based metric learning. We added a better description of the loss in the revised paper.
>
>
> > Q4: The paper lacks a dedicated Conclusion section, with Section 7 not effectively serving as a substitute conclusion.
>
> Thank you for the feedback, we agree and we have updated the paper to add a separate conclusion section (Section 8). Please see the revised paper PDF for the updates.

---

> ### Author Response · Authors · 2023-11-16
> **Response to Reviewer rc5k (2/2)**
>
> > Q5: It seems that the dataset used in the experiment in Section 5.2 is a non-open dataset. The authors should provide more details on the dataset.
>
> Thank you for the suggestion, we have revised the paper to include more details on the dataset, including a better description of the content, ground truth labels, and example types of adversarial attacks. Please see Section 5.2 in the revised paper for the updated details.
>
>
> > Q6: For a same amount of input, does RETSim (Near-Dup) produce more embeddings than RETSim (Partial-Dup)? If yes, why isn’t the speed separately measured in Table 6? (I think RETSim (Partial-Dup) is slower, since it needs to retrieve against a significantly larger set of text embeddings during de-duplication?)
>
> Yes, RETSim (Partial-Dup) will produce more embeddings than RETSim (Near-Dup) because it will return multiple embeddings for longer texts, rather than a single embedding. However, the inference speed is equivalent since both partial and global embeddings are returned in a single inference call to RETSim. Thus, in Table 6, the reported embedding speed of RETSim is the same for both versions.
>
> We did not compare the indexing and retrieval speeds of RETSim since it would depend more on the specific vector search engine used. For large scale experiments, RETSim can be used with an approximate nearest neighbor (ANN) search algorithm and vector index such as USearch or FAISS for efficient indexing and retrieval at the billion or even trillion scale. ANN algorithms scale sublinearly with respect to index size. Thus, although RETSim (Partial-Dup) will produce more embeddings per document, we expect that the indexing and query speed will not be substantially slower.
>
> Thank you for the question, we have revised the paper based on your feedback to clarify why the embedding speed is the same for the two RETSim versions. We also highlighted the fact that RETSim (Partial-Dup) offers a tradeoff between finer-grained deduplication at the cost of producing more embeddings to index and search.
>
> ---
>
> Thank you again for the helpful feedback! Please let us know if you have any additional questions or suggestions, we are more than happy to make further revisions to improve the paper.

---

> > ### Comment · Reviewer_rc5k · 2023-11-17
> >
> > Thank you for your response. The authors' feedback, along with the revisions made to the paper, has effectively addressed most of my questions. I've updated my review accordingly.
> >
> > Additionally, I'd like to propose that the authors emphasize in the paper that the "speed" presented in Table 6 specifically means the embedding/hashing time in the table caption and body text. It is important to clarify that this measurement excludes the indexing & retrieval time from the overall speed calculation.

---

> > > ### Author Response · Authors · 2023-11-17
> > > **Response to Comment by Reviewer rc5k**
> > >
> > > Thank you for updating your review and the additional feedback!
> > >
> > > We complete agree and we have changed the caption in Table 6 to be "Embedding/hashing speed of RETSim vs MinHash + LSH on the Wiki-40B dataset.". We have also updated the body text in Section 5.1 to emphasize that we are comparing only the embedding speed and that indexing/retrieval speed will depend on the specific vector index and search algorithm used.
> > >
> > > Please let us know if you have any other questions or suggestions!

---

### Official Review · Reviewer_pVcX · 2023-11-02

**Soundness:** 4 excellent
**Presentation:** 3 good
**Contribution:** 3 good
**Rating:** 8
**Confidence:** 4

**Summary:**

The paper introduces RETSim (Resilient and Efficient Text Similarity) to identify near duplicate whole text or partial text efficiently, with applications in deduplication, clustering, etc. The proposed model is designed to be lightweight and learned with simple corruption-like augmentations (hence mimicking the typical dedup errors). The proposed method has been carefully tuned and engineered and demonstrated better performance than both non-learned approaches and learned methods.

**Strengths:**

* The problem of dedup is a fundamental one and has many traditional applications. In the era of LLMs, it gains one more important role as deduplicating the training dataset, which is one of the crucial factors to training a good LLM.

* Although the proposed technical solution is simple, it is designed with care (to balance the sophistication and effectiveness) and hyper-parameters are carefully tuned.

* The performance results are excellent over a comprehensive set of tasks and datasets.

**Weaknesses:**

* There is no analysis on why the almost straight-forward design produces better results (F1) than other learned algorithms. From the ablation study (e.g., Sec 6), it seems the benefit is more from tuning parameters?

* Similarly, how important is the training corpus and the data augmentation methods? E.g., what if the data distribution, corruption pattern, etc. on mC4 is drastically different from those in the testing corpus?

* MinHash + LSH achieves a strong performance on real dataset (Table 4). It seems there is no detail on the parameter of the LSH part, and therefore it is hard to know how much performance loss is due to the LSH part (e.g., if the number of "OR" part is small, it may hurt the performance). Also why +LSH? I can understand that MinHash + LSH is a practical solution for large corpus due to its good runtime efficiency, but run-time efficiency is not report for the proposed method. In fact, it is not clear whether and which index you use to find the near duplicate (chunks) with your method. Do you use a vector index and which one?

* It will be better to perform a case study or detailed error analysis to understand the false positive and false negatives for the proposed method.

**Questions:**

See the questions asked in the "Weakness" part.

---

> ### Author Response · Authors · 2023-11-16
> **Response to Reviewer pVcX (1/2)**
>
> We thank the reviewer for their positive feedback and insightful comments, which helped us to improve the paper. Based on your feedback, we have revised the paper to add more details on the baseline algorithms and vector index. We also added an additional section to the Appendix (A.6), where we sampled a set of false positives and false negatives from the NEWS-COPY dataset to further analyze the efficacy of RETSim.
>
> Please find our responses to your questions below and the revised paper for more details.
>
> > Q1: There is no analysis on why the almost straight-forward design produces better results (F1) than other learned algorithms. From the ablation study (e.g., Sec 6), it seems the benefit is more from tuning parameters?
>
> Indeed, the architecture used by RETSim is fairly straightforward. We did try numerous more complex architectures and pre-training methodologies, with minimal to no gain at the expense of significantly slower models. Our best guess as to why we are not able to leverage more advanced models or larger capacity is that the problem of near-duplicate detection and syntactic textual similarity is simple enough to solve with a 2-block transformer architecture. The main difficulty to get high performance was training data generation, where finding the right balance between not enough and too much text augmentation was difficult to achieve. On the other hand, we do not believe that hyperparameter tuning was the reason RETSim outperformed other methods. Some suboptimal hyperparameter settings (e.g. 128-dim embedding size) from the ablation study are still roughly close in performance as the final selected value, whereas RETSim often outperformed baseline algorithms by a significant margin.
>
> The main reason why RETSim surpasses existing neural text embeddings and hash-based algorithms for near-duplicate detection is that it is significantly more resilient to noise such as adversarial attacks and typos (Figure 2). We attribute this improvement to RETSim’s character-level tokenizer which has been shown to be more typo-robust than subword-level tokenizers, and also pre-training the RETSim embedding using text augmentations for adversarial robustness. We understand that this may not be a very satisfactory scientific answer, but empirically, this is what allowed RETSim to achieve state-of-the-art performance on near-duplicate text detection benchmarks while remaining computationally efficient.
>
>
> > Q2: Similarly, how important is the training corpus and the data augmentation methods? E.g., what if the data distribution, corruption pattern, etc. on mC4 is drastically different from those in the testing corpus?
>
>
> We train on a large corpus (Multilingual C4) containing ~27TB of crawled web data and many different text augmentations, which we believe is an important contributing factor for RETSim’s performance on near-duplicate benchmarks especially in the case of noise or typos. The amount of augmentation used in the training dataset was an important design decision that we carefully tuned. We found that too little text augmentation led to not enough generalization, while too much led to poor performance.
>
> Empirically, we find that RETSim is able to generalize well to previously unseen types of noise and data distributions that are different from the training dataset. This is exemplified by its success in the email spam experiment, which consisted of spam emails handcrafted by attackers using targeted attacks and corruption patterns meant to bypass existing spam filters that RETSim was not trained on. The NEWS-COPY dataset is another real-world benchmark based on historical news articles with noise mostly from OCR errors, neither of which RETSim was directly trained on. RETSim is able to outperform both neural-based embeddings and ngram-based algorithms on these benchmarks.

---

> ### Author Response · Authors · 2023-11-16
> **Response to Reviewer pVcX (2/2)**
>
> > Q3: MinHash + LSH achieves a strong performance on real dataset (Table 4). It seems there is no detail on the parameter of the LSH part, and therefore it is hard to know how much performance loss is due to the LSH part (e.g., if the number of "OR" part is small, it may hurt the performance).  Also why +LSH? I can understand that MinHash + LSH is a practical solution for large corpus due to its good runtime efficiency, but run-time efficiency is not report for the proposed method.
>
> Thank you for the suggestion, we agree that this is important to report. The settings for MinHash and other algorithms for this experiment are included in Appendix A.3. For MinHash + LSH, we use 256 hash functions, trigrams, and a Jaccard similarity threshold of 0.5. Using 256 hash functions is recommended by the datasketch library [1] to improve performance over the default 128 hash functions and also commonly used in the literature, so we believe that the performance loss from using LSH should be minimal. We do use MinHash + LSH for computational efficiency, since using MinHash without LSH would have taken significantly longer to run than RETSim and MinHash + LSH (which only takes a few minutes for this dataset).
>
> We have revised the paper to explicitly state the parameter settings for MinHash + LSH in the main paper as well.
>
> [1] MinHash — datasketch 1.6.4 documentation (https://ekzhu.com/datasketch/minhash.html)
>
> > Q4: In fact, it is not clear whether and which index you use to find the near duplicate (chunks) with your method. Do you use a vector index and which one?
>
> Thank you for the suggestion, we have revised the paper to include this information. We use USearch’s default vector index for the Wiki-40B deduplication experiment, which uses HNSW for approximate search (Section 5.1). We use our implementation of exact nearest neighbor search for all other experiments on smaller datasets for embedding models, including RETSim.
>
> > Q5: It will be better to perform a case study or detailed error analysis to understand the false positive and false negatives for the proposed method.
>
> This is an excellent suggestion, we have added a set of sampled false positives and false negatives on the NEWS-COPY deduplication dataset to Appendix A.6 (Tables 16 and 17) as a case study. From manual evaluation, we noticed that false negatives on this dataset seem to be due to texts being heavily corrupted by typos or OCR errors. On the other hand, we noticed that for many false positive examples (Table 17), the texts do appear to be near-duplicates of each other (and RETSim detects them as near-duplicates), although the ground truth label in the dataset does not mark them as near-duplicates. This is an interesting discovery and we have revised the paper to make a note of potentially noisy labels for the NEWS-COPY dataset. Beyond analyzing specific examples, we are uncertain how to do detailed error analysis to explain why the model makes certain mistakes, and we did not find any particular patterns that suggest systematic shortcomings that we can highlight.
>
> Please let us know if you have any additional questions or suggestions. Thank you again for the helpful feedback!

---

### Official Review · Reviewer_XRFc · 2023-11-09

**Soundness:** 3 good
**Presentation:** 4 excellent
**Contribution:** 2 fair
**Rating:** 6
**Confidence:** 4

**Summary:**

This work addresses the problem of learning robust metric embeddings for near-duplicate text retrieval, clustering, and dataset deduplication tasks. Towards this end it proposes a lightweight multilingual deep learning model called RETSim. The work argues that previous solutions for near-duplicate text retrieval are not resilient to typos in the text documents and are vulnerable to adversarial attacks. Further, some of the previous solutions are very parameter-sensitive and consequently need heavy tuning and/or are too big to be used effectively in real-world applications of near-duplicate text retrieval. The proposed solution to the problem of learning robust metric embeddings for near-duplicate text retrieval makes use of a large typo-augmented training corpus, RETVec text vectorizer, a lightweight  transformer block (Hua et al., 2022) and  a metric learning training regime. The tokenizer splits the text into multiple chunks of size at most 512 characters and encodes each chunk using a character encoder. Each encoded chunk is fed to a lighweight transformer that computes a 256 dimensional embedding for the chunk. The embeddings from all the chunks of a text input are then combined into an embedding for the full text by averaging.

The model is trained on a dataset derived from the multilingual C4 dataset by applying sentence, work and character level augmentations on the text. Multi-Similarity Loss is used in training which attempts to make augmented versions of the same text closer to each other in the embedding space.

In addition to addressing the technical problem of measuring similarity of near-duplicate texts, this work introduces a new benchmark for the evaluation of such methods. The benchmark dataset called W4NT3D is particularly oriented towards evaluation of near-duplicate text retrieval models on multilingual text corpora in the presence of typos, word manipulations, and sentence/paragraph-level modifications.

The work presents results from the experiments comparing the proposed method with several baselines including some hash-based methods and some deep-learning-based methods. Results are presented for W4NT3D dataset. The proposed method performs better than all the baselines overall but not by a huge margin (Multilingual E5-Base is very close) and for Chinese and Japanese languages, it performs slightly worse than the best baseline for these languages (Multilingual USE).

The experimental study also reports results on different types of augmentations on the same dataset. Interestingly, all methods seem to do well for paragraph and sentence level augmentations. For word-level augmentations, hashing based methods do worse than other methods and for character-level augmentations, the proposed method fares best.

Further experimental results are provided for two real-world near-duplicate datasets - NEWS-COPY Deduplication dataset and CORE Near-Duplicates dataset. The proposed method gives significantly better results on NEWS-COPY dataset compared to all the baselines. On CORE Near-Duplicates dataset, the method fares marginally better than the hashing-based baselines but no comparison is made with neural-network-based methods.

The work discusses a practical application of the proposed method - spam email clustering. Results show significant improvement in clustering spam emails compared to three baselines.

**Strengths:**

1. A well-designed approach for near-duplicate detection that is light-weight, robust to typos and works on multilingual data.
2. Encouraging improvements in near-duplicate detection over both hashing-based and neural-network-based methods though Multilingual E5-Base seems to be reasonably competitive.
3. A new dataset is made available that can be used for near-duplicate detection studies.

**Weaknesses:**

1. RETSim is generally slower than MinHash and particularly slow on CPU (Table 6).

**Questions:**

1. In the deduplication experiments on Wiki-40B (English), was RETSim trained on this portion of Wiki-40B? If so then could that also be one of the reasons why RETSim is able to detect substantially more duplicates than the baselines?

2. Why isn't Multilingual E5-Base used as a baseline in spam email clustering experiments?

---

> ### Author Response · Authors · 2023-11-16
> **Response to Reviewer XRFc**
>
> We thank the reviewer for their detailed review and helpful comments. We have also revised the paper based on the feedback we received, including making updates to the text to improve clarity and additional experiments.
>
> Please find our responses to your questions and comments below.
>
> > Q1: RETSim is generally slower than MinHash and particularly slow on CPU (Table 6).
>
> Yes, it is true that RETSim is generally slower than MinHash, particularly on CPU. On the other hand, RETSim is much faster and smaller than existing neural-based embeddings such as Multilingual E5-Base, and significantly closes the speed performance gap between hash-based and neural network-based methods. Our current tokenizer implementation is single-threaded and in Python, so we believe there is substantial room for improvement in terms of speed. Overall, RETSim provides a tradeoff between better performance (better near-duplicate detection, increased adversarial robustness) vs speed compared to hash-based algorithms. We strongly believe that RETSim offers a broadly applicable algorithm that has already proven itself on large scale tasks such as spam email filtering, and surpasses state-of-the-art on near-duplicate detection benchmarks from the literature.
>
> > Q2: In the deduplication experiments on Wiki-40B (English), was RETSim trained on this portion of Wiki-40B? If so then could that also be one of the reasons why RETSim is able to detect substantially more duplicates than the baselines?
>
> RETSim is trained only on the Multilingual C4 dataset [1]. We do not fine-tune RETSim on Wiki-40B (or on any other dataset after training). We do expect that Wikipedia data shows up in the Multilingual C4 dataset, although it would represent a very small percentage of the dataset (which is around 27TB of data). Furthermore, RETSim is a very small model (~536k parameters) and we do not repeat examples during training, so we do not believe that the model has the capacity or capability to memorize any Wikipedia training data that would help it on the Wiki-40B benchmark.
>
> In addition, RETSim is capable of generalizing to previously unseen text datasets, as shown in the spam email experiment (Table 7). This experiment uses production email data never seen by the models during training and RETSim’s superior performance on it supports our hypothesis that RETSim has learned a generalizable text embedding that works well across previously-unseen text distributions, languages, and adversarial conditions. This is most likely the reason why RETSim detects substantially more duplicates than baseline algorithms like MinHash, which has been shown to perform poorly in the presence of noise.
>
> [1] Xue et al. “mT5: A Massively Multilingual Pre-trained Text-to-text Transformer.” arxiv.org/abs/2010.11934, 2020.
>
>
> > Q3: Why isn't Multilingual E5-Base used as a baseline in spam email clustering experiments?
>
> The spam email clustering experiment is run on production data and we are only allowed to run vetted models on our internal system. Thus, Multilingual E5-Base is not available for us to use internally, so we are unable to include it as a baseline in the spam email clustering experiments.
>
> Please let us know if you have any additional questions or suggestions for improving the paper. Thank you again for the review!

---

> > ### Comment · Reviewer_XRFc · 2023-11-20
> > **Author response**
> >
> > Thanks for your response. I acknowledge that I've read it.

---

### Author Response · Authors · 2023-11-16
**Summary of Paper Revisions**

We would like to thank all of the reviewers again for their feedback and suggestions, they were very helpful for us to improve the paper. Please see the revised paper for the following updates.

We made two additions to the experiments in the paper, including:
- Three additional baselines for neural embeddings on the CORE Near-Duplicates dataset (Table 4).
- We added examples of false negatives and false positives on the NEWS-COPY benchmark to better understand the limitations of our proposed method (Appendix A.6).

We have made several updates to the text to improve clarity and provide more detailed information, including:
- We have added a separate conclusion section (Section 8).
- We have expanded upon the methodology to provide more detail on the text vectorizer and loss function used (Section 3).
- We have expanded our evaluation methodology to include more information on the vector index used for Wiki-40B dataset deduplication using RETSim (Section 5.1), as well as MinHash + LSH hyperparameters used in Section 4.3.
- We added more details on the email spam clustering dataset used in Section 5.2.

We have also made several smaller edits to improve overall readability and clarity of the paper. Please let us know if you have any additional suggestions or questions, thank you!

---

### Meta-Review · Area_Chair_b5gC · 2023-12-21

**Metareview:**

This paper introduces a lightweight, multilingual deep learning model (named RETSim which stands for Resilient and Efficient Text Similarity) to produce robust embeddings for near-duplicate text retrieval, clustering, and dataset deduplication tasks. The proposed method demonstrated better performance than existing neural and hashing methods. The reviewers generally agree that the proposed method is simple but effective and applies to a variety of tasks, and the paper's experiments are comprehensive. They also pointed out some weaknesses and questions such as the proposed method being slower than MinHash, lack of analysis why the proposed method (while being relatively straightforward) produces better results, and lack of novelty. The authors have responded to the questions and comments properly during the discussion period and reviewers' final scores end up being 6/8. Therefore, I recommend this paper to be accepted, but strongly suggest the authors revise the paper carefully and add new experiments or the clarifications as discussed in the rebuttal period.

**Justification For Why Not Higher Score:**

Please see the summarized weaknesses above.

**Justification For Why Not Lower Score:**

Please see the summarized strengths above.

---

### Decision · Program_Chairs · 2024-01-16

Accept (poster)